# Modeling opinion misperception and the emergence of silence in online social system

**Daniele Vilone** [1,2�] *, **Eugenia Polizzi** [1�]

**1** LABSS (Laboratory of Agent Based Social Simulation), Institute of Cognitive Science and Technology, National Research Council (CNR), Rome, Italy, **2** Grupo Interdisciplinar de Sistemas Complejos (GISC), Departamento de Matemáticas, Universidad Carlos III de Madrid, Leganés, Spain

he These authors contributed equally to this work.
* daniele.vilone@gmail.com

**Data Availability Statement:** The data obtained in the study are all shown in the paper. The codes for accomplishing the numerical simulations are provided as supplementary material. Further

## Abstract

In the last decades an increasing deal of research has investigated the phenomenon of opinion misperception in human communities and, more recently, in social media. Opinion misperception is the wrong evaluation by community's members of the real distribution of opinions or beliefs about a given topic. In this work we explore the mechanisms giving rise to opinion misperception in social media groups, which are larger than physical ones and have peculiar topological features. By means of numerical simulations, we suggest that the structure of connections of such communities plays indeed a role in distorting the perception of the agents about others' beliefs, but it is essentially an indirect effect. Moreover, we show that the main ingredient that generates misperception is a spiral of silence induced by few, well connected and charismatic agents, which rapidly drives the majority of individuals to stay silent without disclosing their true belief, leading minoritarian opinions to appear more widespread throughout the community.

## Introduction

Adherence to group norms and group acceptance are among the most important factors shaping social behavior in humans. The tendency of the individuals to conform to the behaviors, opinions and social norms of the others is an innate trait of human beings, which sometimes can be in contradiction with the intimate beliefs of the individuals. This interplay between individual and social environment can have deep consequences on how opinions form and spread within groups [1, 2].

People's decisions are often guided by what individuals perceive the majority does or approves of [3–5]. A large body of literature in social psychology has provided extensive evidence about how such a perception may not to be accurate and still guide people's behavior. Identifying the mechanisms behind misperception becomes particularly useful in the context of public opinion formation processes as it can help to better understand the conditions under which unpopular opinions consolidate and spread within a community, even in face of a disagreeing majority.

For instance, people may overestimate the share of community supporting their opinions due to "false consensus" bias, which is the tendency to attribute one's own belief (or in this

information may be requested directly from the authors.

**Funding:** Daniele Vilone was partially supported by project SERICS (PE00000014) under the MUR National Recovery and Resilience Plan funded by the European Union NextGenerationEU. Eugenia Polizzi was partially supported by the EU H2020 ICT48 project "Humane AI Net" under contract n 952026. The funders had no role in study design, data collection and analysis, decision to publish, or preparation of the manuscript.

**Competing interests:** The authors have declared that no competing interests exist.

case, one's private opinion) to others [6, 7]. Such a bias can reinforce the public expression of otherwise unpopular opinions, facilitating polarization and radicalization phenomena [8].

Unpopular opinions may also become widespread due to pluralistic ignorance effects, a situation in which the majority of members privately disagree with a certain opinion or attitude, and yet publicly conform to it under the wrong assumption that most others accept it [9, 10]. Similar outcomes can also be induced by the presence of "vocal minorities", i.e. few individuals loudly expressing their minoritarian opinions, and by spiralling mechanisms that lead a disagreeing majority into silence [11]. Despite the empirical evidences of misperceptions about the popularity of certain behaviors and opinions in different real-world settings [10, 12–16], we still lack a clear understanding of how these misperceptions emerge and impact public debates in online social networks. Indeed, by enabling a few people to easily broadcast their voice to millions of others [17, 18], while simultaneously increasing users' exposure to similar opinions [19, 20] social media may provide environments that systematically distort how user perceive certain (potentially unpopular) opinion to be prevalent within the community, with critical consequences at collective level [21–23].

Several findings confirm the importance of vocal minorities in online debates [24–26] and of network structures in favoring the emergence of misperception. For example, a recent study analyzing the communication network structure of millions of COVID-related fake news on Twitter provides evidence of an overexposure of a large group of passive users to the tweets of a few active users responsible for fabricating most of the contents available online. Such environment can increase the chance that misinformation spreads due to users conforming to the "voice of few", wrongly perceived as representative of the "opinion of many" [27]. The utility of such observational data is however limited by the fact that they do not allow to identify the contribution to misperception due to a potential disalignment between what users chose to publicly declare (that is, their "expressed" opinion) with respect to their non observable, private beliefs (that is, their "private" opinion). Understanding the dynamics of misperception only by means of empirical data is not trivial [28, 29]. In this respect, agent-based modeling (ABM) may represent a useful approach, as it allows building simplified models of the dynamic under investigation along with an *a priori* definition of key variables (e.g., agents' decision-making rules) and system parameters (e.g., network structure) and to observe which macroscopic patterns can spontaneously emerge [30]. Specifically, ABM consists in simulating social dynamics by means of virtual agents that interact among themselves following some established rules [31–33].

Recent effort has been addressed towards modeling specific opinion dynamic processes, in particular the spiral of silence, by incorporating the potential for a disconnection between agents' expressed and private opinions [34, 35] and, in some cases, network heterogeneity [34, 36, 37]. While in principle these works are adequately modelling opinion dynamics for specific scenarios (e.g., elections with major candidates, where binary options are equally likely to occur) they are less suited to examine the processes by which vocal minorities can sway the perception of public opinion toward their side (see Refs. [38–40] for some relevant theoretical works about contrarian minorities swaying public opinion). Simulations have also been used to explore whether opinion misperception can emerge simply due to the geometry of the network, showing how simple topological effects can increase the perceived prevalence of few high-degree nodes in the eyes of their less connected neighbors: in the context of online systems this "illusion of majority" bias, rooted in the network structure, has indeed been suggested to act as an exogenous (that is, not depending on the agents' behavior) mechanism facilitating the spread of unpopular opinions and behaviors [41]. Yet, this crucial point requires still to be empirically tested. Finally, simulation work exploring biases in agents' inference processes (e.g., inferring others' beliefs through their observable behavior) provides

support for the role of "endogenous" sources of misperception (i.e., that depend on agents' internal decision making system) in driving the spread of unpopular norms in a system [42]. Models of this kind could also be applied to investigate how the voice of minorities can influence public discourses in systems characterized a potential disallignment between individuals' expressed and private opinions.

New models and theoretical approaches that account for both exogenous and endogenous determinants of opinion misperception are thus necessary. In this study we will address such a challenge by modeling communication among agents in a prototypical online community scenario: when a community of agents agreeing (to varying degrees) on a certain popular opinion, is joined by a handful of agents "loudly" expressing their disagreeing opinion. For example, we can think about the exordia of the antivax movement, when few activists started to spread in internet their skepticism among communities which until their appearance had never questioned the utility of vaccinations [43, 44]. We will refer to these agents as "hard-core minority", because of the similarity with the so-called "hard cores" in the definition by Noelle-Neumann and Matthes & colleagues [11, 45] that is, people that stick to their opinion regardless of the prevailing opinion climate. In our model, hard-core minorities will thus be characterized by having an expressed opinion always in line with their privately held one). In the following sections we will thus explore whether such a minority can distort the perceived public opinion and the repercussion of misperception at the system level.

## Exogenous contributions to opinion misperception

In this section we investigate whether a substantial distortion in the opinion distribution, as perceived by agents, can emerge mainly due to the complexity of the network, without resorting to the cognitive features of the agents and their interaction. More precisely, it has been pointed out that in complex networks the most connected nodes may be over-represented in the neighborhood of other less connected nodes, and that such a condition can lead the former ones (and so their observable features) to be perceived as much more prevalent compared to their actual distribution within the larger population (i.e., "illusion of majority"). This effect is due to the so-called "friendship paradox" [46–48], which states that in a heterogeneous topology the average degree of the neighbors of an agent is higher than the network's average degree itself. Building on previous modeling work [41] we thus create a model where agents have minimal cognitive features and publicly express their opinions in a communication network characterized by a clear majority-minority belief (or "private" opinion) distribution. By manipulating the popularity of hard-core minoritarian agents we aim at evaluating the weight of topological heterogeneity in driving the dynamics of opinion misperception at macro-scale level.

### The model

We consider a system of $N$ agents arranged on a given network, defined by its adjacency matrix $\hat{M}_{ij}$ and characterized by its degree distribution $P(x)$ [47]. As already stated, the internal structure of the agents and their interaction rules will be outlined in the simplest possible way. Specifically, each agent $i$ is defined as follows:

- The belief, or private opinion, $b_i$, which can assume one of two possible values, $b_i = \pm 1$. As done in previous work (e.g., [34, 35, 42]), for simplicity we model agent's belief as fixed in time. From a theoretical perspective, such a choice is justified by the fact that the primary focus of these models (and of the theories behind, e.g., [9, 11]) is about explaining the process that leads individuals to publicly express opinions that do not align with their private beliefs,

and not about changes in beliefs. Accordingly, the agents in our model do not change what they privately think, but only what they publicly express. In this model, "belief" and "private opinion" are thus meant as overlapping terms.

- The strength of the belief $\sigma_i \in (0, 1]$, also constant in time, which specifies how strongly the agent holds its belief: when requested to publicly express its opinion, the agent will answer its true belief with probability $\sigma_i$.

- The expressed opinion $\omega_i$, which may or may not be equal to the agent's private belief depending on the probability $\sigma_i$. At the initial stage of the dynamic (e.g., when no interaction among agents has occurred yet), it may also assume the 0 value besides ±1.

Once set the agents in the network and assigned the initial values of each variable (see below), the dynamics takes place. The elementary time-step of the dynamics is defined as follows:

- An agent $i$ is selected at random;

- With probability $\sigma_i$, the agent will express an opinion equal to its private one, therefore $\omega_i = b_i$;

- With probability $1 - \sigma_i$ the agent will express the opinion of the majority of its own neighbors:

$$\omega_i = \text{sign} \left[ \sum_{j \in \Gamma_i} \omega_j \right] , \tag{1}$$

where $\Gamma_i$ is the set of the nearest neighbors of the agent $i$ (if the result is zero the expressed opinion will be +1 or −1 with equal probability), the sign function gives back +1 (−1) if its argument is positive (negative), sign(0) = 0;

- A time unit of the simulation will be given by $N$ of these elementary processes;

- In order to have enough statistics, all the results presented in the following are averaged over 10000 independent realizations.

Agents thus undergo a basic social influence process recalling a majority rule [49–51]. In this case however, estimation of majority can only be based on what is directly observable by agents, i.e. the opinions expressed by their neighbours. Neither the global distribution of expressed opinion nor anyone else's private beliefs is known. The adoption of such a heuristic is justified under a bounded rationality view of decision making, e.g., agents acting under limited information and limited capacity to make decisions. Therefore, the majoritarian opinion expressed by agents' neighbors is what agents can use to infer the general "opinion climate", and will thus also correspond to agent's perception of the global distribution of private beliefs.

**Topology.** We considered $N$ = 5000 agents on a Scale Free (SF) network with degree distribution with exponent $\lambda$ = 2.2, which characterizes with a good approximation the topological structure of many social media communities [46, 52–54], and in particular on Twitter [55]. Therefore, we consider a network whose degree distribution follows the distribution below:

$$P(k) \propto k^{-\lambda} , \tag{2}$$

where $\lambda$ = 2.2. We generated such networks for our simulations by means of the Molloy-Reed algorithm [56], so that for numerical reasons the minimum allowed degree is 2, the maximum $\lfloor \sqrt{N} \rfloor$ [57].

**State variables.**   The first important variable describing the state of the system is $\langle\omega\rangle$, that is, the average expressed opinion:

$$\langle\omega\rangle = \langle\omega\rangle(t) \equiv \frac{\sum_{i=1}^{N} \omega_i(t)}{N} \ .$$

Analogously, we consider the average belief $\langle b\rangle$, defined in the same way as $\langle\omega\rangle$, which does not change over time. In order to evaluate the extent to which network features contribute to bias agents' perception of others' opinions we also include a variable measuring the opinion climate that an agent perceives in its immediate surrounding. In order to define such variable, we start from the neighbors' average expressed opinion $\Delta_i$ of each agent:

$$\Delta_i = \frac{\sum_{j\in\Gamma_i}\omega_j}{|\Gamma_i|} \ ,$$

where $|\Gamma_i|$ represents the degree of the node $i$. Therefore, we define the average local opinion climate $r(t)$ as

$$r(t) \equiv \langle\Delta_i(t)\rangle_i = \frac{\sum_{i=1}^{N}\Delta_i(t)}{N} \ .$$

Notice that all these quantities are defined in the real interval $[-1, +1]$.

As said, agents can only observe what is publicly expressed in their immediate surrounding, which may or may not be representative of the global distribution, and may or may not be aligned with the distribution of private beliefs. The potential discrepancy between the average opinion expressed locally and what is expressed globally will capture the contribution of topology to opinion misperception. Similarly, we can use the mismatch between the average opinion expressed around them and the true distribution of non observable beliefs to measure the full extent of misperception. In Table 1 we provide a summary of the meaning of the main variables used to describe the state of the system.

**Initial conditions.**   The initial settings are chosen in order to mimic a prototypical social media environment characterized by a large consensus around a certain topic (that is, a majoritarian opinion) with only few users voicing out their (minoritarian) opinion: as already anticipated, we will refer to these latter agents as "hard-core minority" (or simply HC agents) because they a) -privately- hold an unpopular opinion, and b) -publicly- stick to such opinion regardless of the prevailing opinion climate ([11, 45]). We thus assign to our HC agents a belief $b = -1$ and maximum strength, (i.e., $\sigma = 1$) that is, the opinion expressed is always in line with their private one. Crucially, such agents are selectively placed on nodes with degrees $K_c$, in order to highlight the role of the network's topology. In every other node with degree $k \neq K_c$ the remaining agents (majority) will hold belief $b = +1$ and strength $\sigma$ picked up from a uniform random distribution between 0 and 1. It follows that the distribution of HC agents will

**Table 1. List of the quantities utilized for the system's description.**

| Quantity | Definition |
|---|---|
| $\langle b\rangle$ | Average private opinion (belief) |
| $\langle\omega\rangle$ | Average expressed opinion |
| $r(t) = \langle\Delta_i\rangle$ | Local opinion climate $\equiv$ neighbors' expressed opinion |
| $\langle b\rangle - \langle\omega\rangle$ | Private vs Public Mismatch |
| $\langle b\rangle - r(t)$ | Misperception |
| $r(t) - \langle\omega\rangle$ | Topological contribution to misperception |

be given by the same $P(k)$ given in Eq (2), that is, $P(K_c) \propto K_c^{-\lambda}$. Specifically, when $K_c$ is high the network will be composed by a very small minority of HC agents, but very well connected. Contrarily, when $K_c$ is low HC agents will be relatively more abundant but also less connected. We will thus study how the model behave by systematically varying $K_c$, the degree of nodes (and so, how central they are in the network) on which the HC minority is placed. Once set the initial distribution $\{b_i\}$, the first time an agent is required to express its opinion, if no one of its neighbors has already expressed it, with probability $1 - \sigma_i$ the agent will express a null opinion: $\omega_i = 0$.

## Results

In Fig 1(top) we show the final values of $\langle b \rangle$ (private belief), $\langle \omega \rangle$ (expressed opinion) and $r$ (local opinion climate) as functions of HC agents' degree. As it can be observed, the average belief is always quite high because of the specific distribution of agents in the system. For example, the prevalence of HC agents reaches $\simeq 18\%$ at $K_c = 3$, while it decreases to $< 0.001\%$

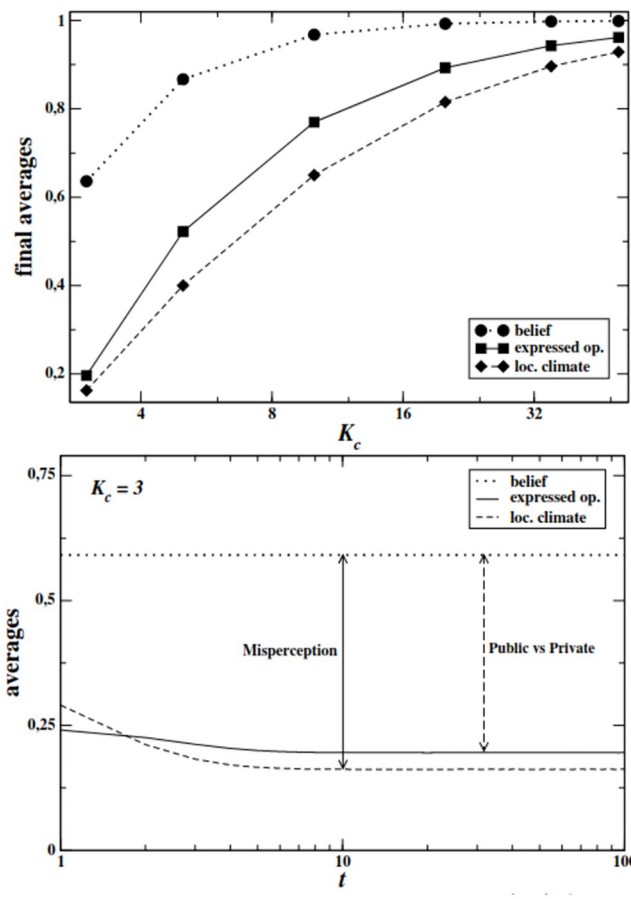

**Fig 1. Top:** Final values of the average expressed opinion $\langle \omega_f \rangle$ (continuous line), average local opinion climate $\langle r_f \rangle$ (dashed line), and average belief $\langle b \rangle$ (dotted line) as functions of HC agents' degree $K_c$. Scale-free network, size $N = 5000$, $\lambda = 2.2$, averages over 10000 independent realizations. **Bottom:** Time evolution of the average expressed opinion $\langle \omega \rangle(t)$ (continuous line), local opinion climate $r(t)$ (dashed line), and average private belief $\langle b \rangle$ (dotted line, constant in time) for $K_c = 3$. Scale-free network, size $N = 5000$, $\lambda = 2.2$, averages over 10000 independent realizations. We highlighted the contributions to misperception, i.e. the difference between the average belief and the average local climate (continuous double arrow), and the discrepancy between what agents privately believe and what they publicly express (dashed double arrow).

at $K_c = 50$, according to the Eq (2). In all $K_c$ cases, the average expressed opinion is always lower than the average belief, meaning that many agents privately holding the majoritarian opinion end up expressing the opposite one. Crucially, the local opinion climate, represented by $r$, is always lower than the opinion expressed on average by the agents in the population, suggesting an over-exposure to minority opinion in the local neighborhood. However, such a distortion is rather small (at most around 15% for $K_c = 10$). When taking into account the actual distribution of private beliefs, the discrepancy becomes considerably larger. This is easily observable in the $K_c = 3$ scenario shown in Fig 1(down): agents observe an average opinion around them of slightly less than 0.2, while the average private belief is around 0.67. Overall, the behavior of this model suggests that topology itself contributes only partially to opinion misperception and that most of the phenomenon is due to the mismatch between agents' expressed and private opinions. Such a pattern implies that even when agents are modeled with minimal cognitive ingredients (i.e., strength of belief), the main source of bias appears to be endogenous (that is, mainly due to agents' behavior) rather than exogenous (that is, their position in the network). Finally, with increasing $K_c$ all the curves tend to 1 because the size of the minority becomes drastically small, giving rise to a saturation effect, as shown in Fig 1 (top).

## Endogenous contributions to misperception

The results obtained in the previous section show that topological effects alone are not enough to explain the overall misperception of the opinions held among the population. In order to better understand what drives the observed dynamics, we thus proceed by refining the characterization of the endogenous component of opinion misperception by acting on agents' interaction rules. Implementing complex agents' socio-cognitive features in ABM is a non-trivial task [58], mainly because of the limits of encompassing the complexity of human cognition in a relatively simple algorithm. There are many ways to model the internal dynamics of agents, as for example refining models from Opinion Dynamics theory [59]. A useful example comes from the study of Merdes [42] which investigated the role of belief misperception in the emergence of unpopular norms by means of simulations. In that model, agents have fixed, non observable, beliefs and are subject to social pressure from the surrounding neighborhood with regard to which behavior to adopt (e.g., to follow or not the norm by conforming to the most prevalent behavior). Agents' choice is driven by the effort to minimize the gap between their social expectations—what agents believe most others do (or perceive so)—and their own, potentially disagreeing, private belief. Such a characterization of agents' decision rule, along with the possibility for them to behave differently from their private beliefs creates the conditions for the emergence of misperception at macro-scale level, i.e., most agents behave in a way which is not aligned with what would be expected from the underlying distribution of private preferences in the population. Norms can be thought as a specific kind of opinion, and our differentiation between agents' private and publicly expressed opinion (observable choice) allows us to adapt their formulation of social influence to describe the endogenous contribution to opinion misperception. Accordingly, in our model the social component will be formulated as the effort by the agents to minimize the gap between what they perceive is the majoritarian opinion around them and their own private—and potentially disagreeing—belief. As in [42], such assumption should be able to capture human tendency to reduce as possible the deviations from others' behaviors, norms and opinions [4]. Crucially, people's perception of their social surrounding (whether in line or not with their private belief) may not only affect which opinion to express, but also, whether to express it at all. More precisely, the perceived disagreement with the opinion mostly expressed in the agents' neighborhood is expected to limit

agents likelihood to express their own private belief [11] and may thus further contribute to the emergence of opinion misperception in the system. Accordingly, we incorporate this insight by adding the possibility for agents to stay silent.

## The model

In order to describe the above processes mathematically, a specific *goal function* $G_i$ is introduced for every agent. This function gives the payoff gained by the agent according to its own and neighbors' state, so that the payoff increases when the state of the agent is closer to the others', but decreases when the agent is forced to change its own previous state. In other words, agents tend to minimize the disagreement with their neighbors' opinions as well as the cost of publicly expressing an opinion in contrast with ones' own private belief. More precisely, we assume that agents have a belief, or private opinion, $b_i = \pm 1$, constant in time, and an expressed opinion $\omega_i = 0, \pm 1$, where $\omega_i = 0$ critically adds a preference for staying silent (in the first model, the option $\omega_i = 0$ can be adopted only initially, e.g., when no neighbors have yet expressed themselves: therefore, it is not meant to represent agents' choice to stay silent). Again, to each agent $i$ is attached a belief's strength $\sigma_i$, which can assume a real value from 0 (no strength at all), to 1 (maximum of confidence in own belief). The dynamics in this case proceeds as follows:

- An agent $i$ is selected at random;

- To decide what to express or whether to remain silent, the agent $i$ maximizes the *goal function* $G(\omega_i)$ defined as

$$G(\omega_i) \equiv - \left| \frac{\sigma_i b_i + \sum_{j \in \Gamma_i} \mathcal{A}_i^j \, \omega_j}{|\Gamma_i| + 1} - \omega_i \right| \, , \tag{3}$$

where $|\Gamma_i| \equiv \sum_{j \in \Gamma_i} |\omega_j|$, $\sigma_i$ is the strength of the belief of player $i$, and the function $\mathcal{A}_i^j$ is the "influence" of the agent $j$ with respect to agent $i$ itself (that is, how much the expressed opinion by $j$ influences agent $i$'s behavior): we set $\mathcal{A}_i^j = \frac{\deg(j)}{\deg(i)}$ to incorporate the fact that more charismatic individuals tend to get more connections with others than less charismatic ones [60, 61];

- Once found the value $\bar{\omega}_i$ which maximizes $G(\omega_i)$, such $\bar{\omega}_i$ will be the opinion expressed by $i$ in the next interaction (if $\bar{\omega}_i = 0$, agent $i$ stays silent).

- Notice that one could equivalently define the cost function $C(\omega_i) \equiv -G(\omega_i)$, and proceed in the same way with the only difference being that $C$ has to be minimized.

**Topology.** As for the first model, we analyze Scale-Free networks with degree distribution $P_{sf}(k) \propto k^{-\lambda}$ with $\lambda = 2.2$, as in Eq (2), generated again by means of a Molloy-Reed algorithm (see Sec.); we also accomplished some simulations with $\lambda = 1.5$ in order to test the effect of increasing heterogeneity.

**State variables.** To quantitatively characterize the model's behavior, we use the same variables defined in Subsec., that is, the average expressed opinion $\langle \omega \rangle$ and the average perception $r$ through the local opinion perception $\Delta_i$.

**Initial conditions.** As for the first model, we consider systems with most agents having $b_i = 1$ and few HC agents with $b_i = -1$ put on nodes of degree $K_c$; we set $\sigma_i = 1$ for the latter agents, and uniformly distribute it in the interval [0, 1] for the former ones. Finally, we always assume $\omega_i = 0 \; \forall i$ at the very beginning of the dynamics.

As in the previous section, we will study how the model behaves by systematically varying $K_c$, the degree of nodes on which HC agents are placed, and all the results presented in the following section are averaged over thousands independent realizations.

## Results

In Fig 2 (top) we show the temporal evolution of agents' average expressed opinion $\langle\omega\rangle$ for systems on SF networks ($\lambda$ = 2.2, $N$ = 5000), for three different exemplifying $K_c$ scenarios. Again, increasing $K_c$ values correspond to systems with fewer but better connected (and so, more "charismatic") minoritarian agents.

The average opinion expressed by agents is quite close to 1 at the initial stages of the dynamics (despite the fact that $\omega_i(t = 0) \equiv 0 \,\forall i$, the agents express an opinion in line with their private belief the first time they are asked to), but decreases up to a relatively lower final value with similar qualitative behavior for different $K_c$. Also, the final values of expressed opinion are higher with increasing $K_c$, as the overall size of minority in the systems becomes drastically small (e.g., at $K_c$ =20 only 3 out of 1000 are HC agents), but with a saturation effect for larger $K_c$.

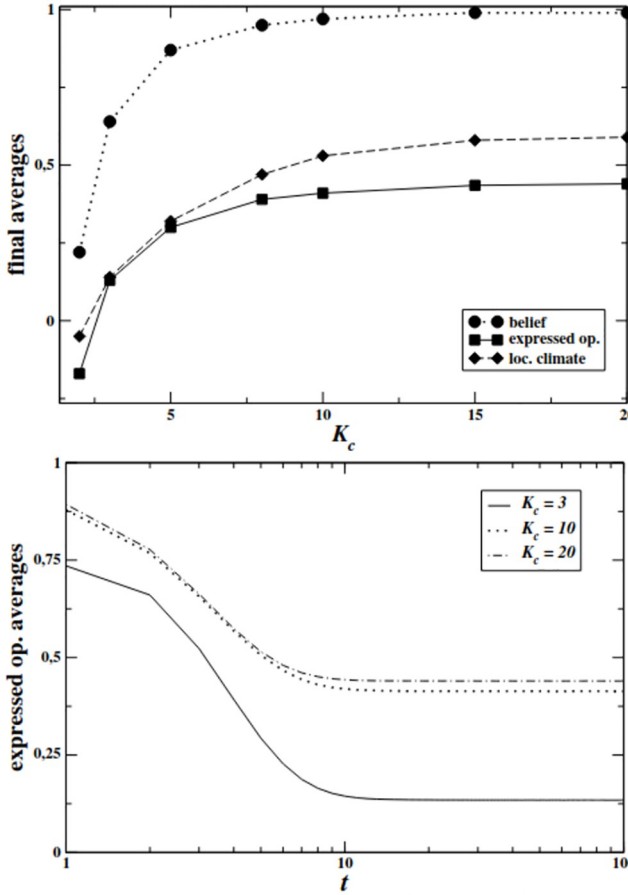

**Fig 2. Top:** Behavior of the final average belief (dotted line), expressed opinion (continuous line) and local opinion climate (dashed line) as functions of the HC agents' degree $K_c$ for a highly heterogeneous SF networks ($\lambda$ = 2.2). System size $N$ = 5000, results averaged over 5000 independent realizations. **Bottom:** Time evolution of the average expressed opinion for systems on SF networks with $\lambda$ = 2.2, and three different values of $K_c$. System size $N$ = 5000, averages over 5000 independent realizations.

In Fig 2(top) we show the final values of the average belief, local opinion climate and expressed opinion, as functions of $K_c$. The average belief value (which is constant in time) increases with $K_c$ because the number of HC agents decreases according to the increase in their degree. However, the average opinion expressed by agents, both as it is perceived locally as well as it is expressed globally, is always lower than the belief value in the population, and a stable plateau is reached with increasing $K_c$ values. This pattern is very different from the results of first model, where the average value of expressed opinion approached agents' belief values as $K_c$ increases (see Fig 1 (bottom)). Such a finding suggests that, even when being of a negligible number, well-connected minorities are able to sway the global opinion towards their side, despite the fact that almost every agent holds a disagreeing private belief. In order to better understand the observed pattern, we look at how the proportions of agents choosing to express the majoritarian or minoritarian opinion (or to stay silent) change according to $K_c$.

Fig 3 shows the proportion of agents based on their expressed opinion $\omega$ at the end of dynamics for several values of $K_c$ from 2 to 20 as compared to the fixed distribution of their private beliefs. The figure suggests that the shift in public opinion is not due to majoritarian agents publicly aligning to the opinion of the minority, but to majority's choice of staying silent. Indeed, silence becomes the most effective solution for agents to minimize the cost of having a different belief from their (even though misperceived) immediate social surrounding without the need to deviate from their inner views. From a dynamical point of view, the possibility to avoid expressing opinions induces a cascade where agents surrounded by neighbors expressing mostly the opposite opinion or being silent end up staying silent themselves, driving others to stay silent on their turn, and so on. Such a dynamics leads to a final state where the average expressed opinion is much more skewed toward the minority compared to the average group's private beliefs. Interestingly, around $55 \div 60\%$ of subjects (almost everyone of which holding positive beliefs) end up being silent, regardless of $K_c$. In order to shed light on

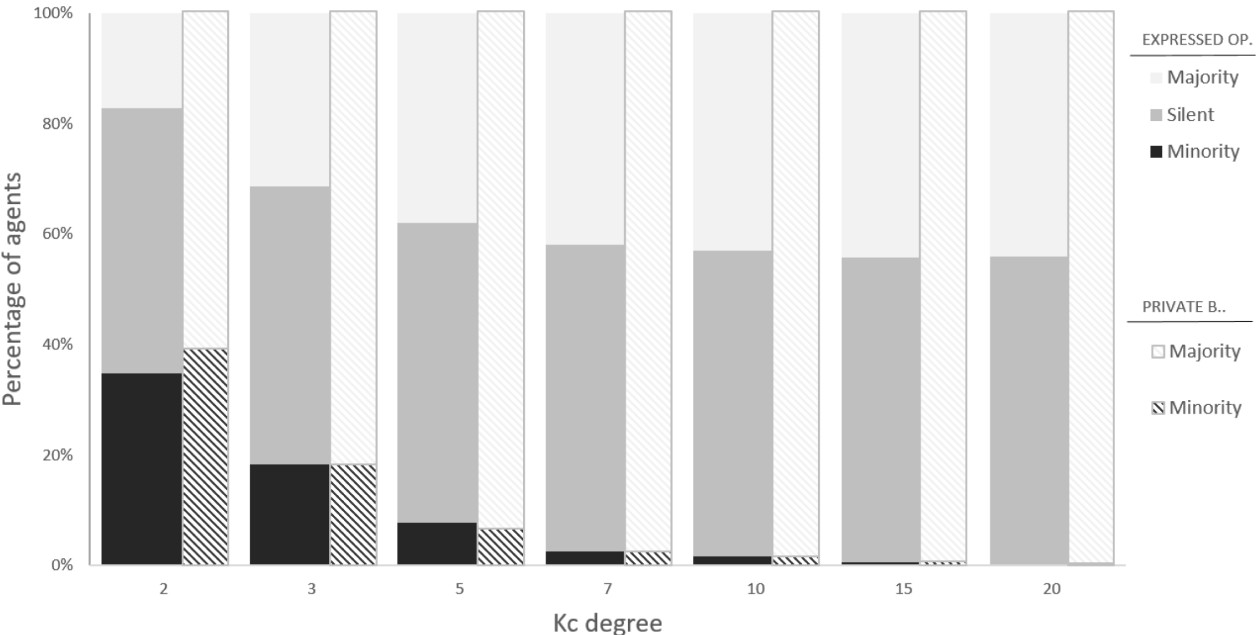

**Fig 3. Histogram showing the fractions of agents expressing the majority/minority opinion or staying silent (full bars) with respect to the true fractions of majority-minority agents in terms of private beliefs (shaded bars), for different degree $K_c$ of hard-core minority.** Scale-free networks, size $N = 5000$, $\lambda = 2.2$.

this effect, we compute the number of silenced agents per HC agent and plot it as a function of their degree $K_c$. Fig 4 shows a superlinear relationship between the HC agents degree value and the amount of silenced agents. Specifically, when $K_c$ is low, e.g., 2, the size of the HC minority is large (around 35%), but every HC agent drives less than 2 agents to silence. This suggests that for low $K_c$ silence mostly arises due to short-range dynamics, e.g., many HC agents directly influencing their immediate neighbors, and very few others. At $K_c = 20$, the minority is negligible (<0.1%), but for every HC agent there are $\sim$ 150 agents that get silenced, much more than their neighbors. Such a pattern suggests the presence of long-range dynamics, as silence spreads even beyond HC agents' direct connections. Therefore, for high $K_c$ the increasing centrality of few well-connected HC agents balances the reduction in their number, inducing the saturation effect that stabilizes the opinion expressed (and its local perception, too) toward lower values. Such a saturation effect is also reflected in the exponent of the superlinear relationship between HC agents and silenced agents, which is very close to the network $\lambda$ value: since the overall fraction of silent individuals remains approximately constant while the number of HC agents decrease as $K_c^{-\lambda}$, then the amount of the former versus the latter must increase with an exponent very close to $\lambda$ itself.

As in the first model, the direct effect of topology in distorting agents' perception of others' opinion is again rather small. Interestingly, and in contrast to the former, the opinion that users can observe locally is actually closer to the true value of private beliefs as compared to the opinion expressed on average by agents. Such an effect is possibly due to the stochasticity of the model, e.g., by few agents holding positive beliefs randomly ending on nodes with degree values even higher than the ones held by HC agents. Such agents are thus able to limit the "traction" of a well connected HC minority, whose effect is therefore top-down. Such a finding suggests that the direct topological effect on opinion misperception is, besides small, also non-universal. Finally, we replicated the model on a Scale-free network with $\lambda = 1.5$, corresponding to a more heterogeneous system than a Twitter-like one. From a qualitative point of view, a similar behavior can be observed (see Fig 5), confirming that the results obtained are stable in the framework of SF networks.

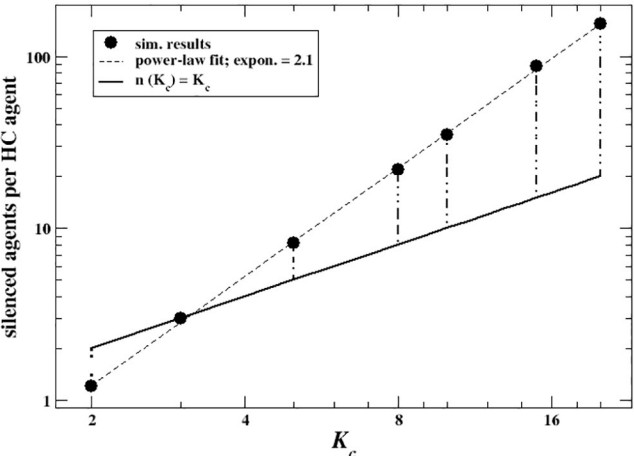

**Fig 4. Black circles:** number of silenced agents per HC agent as a function of $K_c$. **Dashed line:** power-law fit $\sim K_c^v$, $v \simeq 2.1$ (superlinear behavior). **Vertical lines:** difference between the number of agents silenced per HC agent and the actual number of HC agents' nearest neighbors. For $K_c = 2$ the line is dotted because each agent belonging to the minority drives less agents to silence than its number of nearest neighbors. **System parameters:** SF network with exponent $\lambda = 2.2$, system size $N = 5000$; averages over 5000 independent realizations.

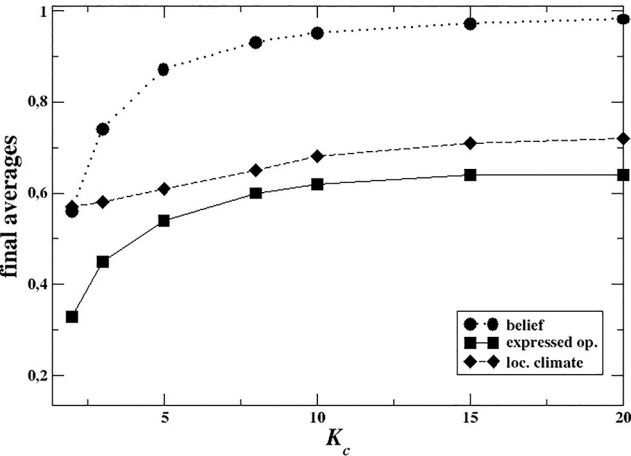

**Fig 5. Final values of the average expressed opinion $\langle \omega_f \rangle$ (continuous line), average local opinion climate $\langle r_f \rangle$ (dashed line), and average belief $\langle b \rangle$ (dotted line) as functions of HC agents' degree $K_c$.** Scale-free network, size $N = 5000$, $\lambda = 1.5$, averages over 5000 independent realizations.

## Discussion

There is increasing evidence that social media communication features increase the potential for vocal minorities to reach a much larger audience than in offline contexts, and to easily give rise to viral phenomena [62, 63]. While in principle such effect may not be deleterious per se, an over-representation of otherwise minority opinions may become problematic if it increases the risk of inflating the support for socially questionable or even harmful views [27, 34]. This study thus aims at better understanding how opinion misperception emerges as a results of agents dynamically interacting in networks with social-media like features. Specifically, we focused on evaluating the contribution of both exogenous (e.g., the structure of connections) and endogenous (e.g., agents' socio-cognitive features) mechanisms allowing few users loudly expressing their unpopular opinions to be over-represented in the eyes of other users and the consequences of such distortion on the dynamics of opinion at collective level. Topological contribution was analyzed by selectively placing such "hard-core" minoritarian agents on nodes with degree $K_c$ and by running simulations for different degree $K_c$ values. Simulation results show that local network effects (in particular, the heterogeneity of the degree distribution) only marginally distort the perceived prevalence of minority opinions with respect to what agents express on average. Besides being marginal, the effect of such topologically-induced misperception also does not seem to be universal. Indeed, under some conditions the perceived prevalence of minority opinions in the local surrounding can even get closer to the actual distribution of agents' private beliefs, thus partially correcting the effects of opinion misperception on the system. Previous modeling effort has shown how "illusion of majority" effects arising simply due to network features [41] can increase the perceived prevalence of minority opinions and act as network based explanation for the emergence of misperception. However, the actual impact of such distortions on collective behavior remains untested. By allowing agents to dynamically interact and by incorporating a differentiation between expressed vs private opinion we can explicitly test the effect of such distortions on public opinion. Our results suggest that agents behavior (e.g., what opinion they decide to express) is driven by them conforming under a wrong estimation of how widely the minoritarian opinion is truly held, which eventually shifts the global public opinion toward such side. Crucially, the

result of the second model suggests that this effect is not due to a majority choosing to align with the minority, but to a majority of agents choosing to stay silent.

A spiral of silence emerges as a consequence of such individual choices. Silence becomes thus a crucial element to explain the observed opinion dynamic patterns. While similar mechanisms have already been noticed and discussed in literature [11, 64, 65], their existence in onlin social media has been subject of debate [64, 66–68]. Observing spiralling processes and their effects in such environments is difficult, mainly because only the opinions of those who publicly express themselves are evident there. Such a barrier can be lifted by means of computer simulations that allow to explicitly model the discrepancies between private and publicly expressed opinions. Although based on a simplistic description of opinion formation, our work suggests that the main feature of social media networks (i.e., the heterogeneity of connections) can promote social biases and contribute to amplify the voice of a motivated few by pushing into silence a disagreeing majority. Surprisingly, the silencing process emerges regardless of the network position or size of minoritarian agents. The present work underlines the importance of network characteristics when it comes to observing the dynamical process. Indeed, we show that when our HC agents are located on low-degree nodes (with fewer direct connections), their lack in centrality (or individual "charisma") is compensated by their larger number, which allow silencing process still to emerge at large scale. On the other hand, when placed on high-degree nodes, their numbers are few, but silence emerges due to their strong connections, thus compensating for the significant reduction in their numbers. Consequently, these two effects balance each other, resulting in a consistently high fraction of silenced individuals. These findings align with the research conducted by Ross and colleagues [34], who explored the impact of social bots (artificial, automated accounts impersonating humans) on diverting public discussions in social networks using a different model. Similarly to our findings, they show that the influence of these bots is not significantly affected by changes in their position or overall number. Notably, in both studies the percentage of silenced individuals reached levels around 50% to 60%. Despite differences in the models and simulation algorithms employed, this agreement suggests that the presented findings may reflect an inherent characteristic of these types of social interactions.

As a consequence of the silencing process the proportion of agents expressing majority and minority opinions changes considerably with respect to the true distribution of private beliefs. However, in most of the investigated scenarios this process does not translate into a complete majority-minority overturn. If we think of opinion formation and spread in social media, this suggests that in case of widely accepted views, small vocal minorities can weaken the strength of majoritarian opinions but this may not be enough to make them dominate public debate, as it has already been pointed out by Acerbi *et al.* [69, 70]. In our model, the only case of inversion in majority-minority proportions is found when the minority exceeds the percentage of 30%. Interestingly, such a percentage is close to what indicated by Centola and Baronchelli [71] as being a tipping point beyond which a minority manages to spread globally into a population of agents. Although our study does not allow (nor it was meant to) to investigate the effects of tipping points on opinion spread, further work could explore the link, i.e., whether opinion misperception can facilitate or hinder the emergence of social tipping points, and the role played by the network itself (e.g., the number and centrality of the agents involved).

In summary, our work contributes to the current literature on opinion dynamics by exploring the determinants of opinion misperception in heterogeneous network systems. We tackle this issue by incorporating insights from socio-psychological literature to model the discrepancy between expressed and private opinions and the resulting effects at large-scale level. Modeling exercise requires often some level of simplification. In our case agents representation was chosen so to allow for a cleaner examination of the dynamics of opinion misperception,

but may not be able to fully capture all aspects of real-world opinion formation and change. For example, implementing agents' private opinions as fixed beliefs is a simplification that, while justifiable, may lead to observe more extreme dynamics than when beliefs are allowed to change over time. Similarly, while we rely on relevant social psychological theories to chose how to model social influence, implementing different rules may lead to different macroscopic outcomes than the one we have observed. In order to understand whether the observed dynamics are translatable to "real" online scenarios the model should also be validated with empirical data. Observation of real networks, surveys and experiments providing data about users' potential disallignment betweem private and expressed opinions could generate valuable insights and empirical evidence to validate and expand our understanding of the processes behind opinion formation, misperception, and their dynamics at different levels.

There are some natural extensions to the model that could be explored in future research. While the two-option model represents a good approximation of a simple scenario involving polarizing debates in social media, (as for instance pro/anti vaccine cases), future work may explore how the dynamics are influenced by the granularity of the opinion scale, the number of levels or categories it contains. For example, when opinions can be expressed at multiple levels individuals may find it easier to identify common ground or moderate positions, and stronger evidence or more substantial differences with respect to their social surroundings could be required to start a silencing process. Extensions could also be developed to explore how misperception dynamics can be shaped by allowing multiple opinions to be logically inter-related. Opinions interdependence is indeed frequently encountered in real life, but still little theoretically developed and empirically explored [72]. Finally, it has been shown that the effect of misperceptions can be alleviated by informing people about the real distribution of others' beliefs [73]. If, as suggested here and elsewhere [27], such biases occur in communication networks too, simulation models could be further used to test interventions to counteract the effects of misperception on public debates, for example by testing whether revealing over-exposure to minority opinions can limit the emergence of silencing processes at system level. Multidisciplinary research combining experimental social science and computational methods can prove to be effective in enhancing our comprehension of the complex dynamics at play in online social systems and help guide future research and interventions in this domain.

## Author Contributions

**Conceptualization:** Daniele Vilone, Eugenia Polizzi.

**Data curation:** Daniele Vilone, Eugenia Polizzi.

**Formal analysis:** Daniele Vilone, Eugenia Polizzi.

**Investigation:** Daniele Vilone, Eugenia Polizzi.

**Methodology:** Daniele Vilone, Eugenia Polizzi.

**Software:** Daniele Vilone.

**Supervision:** Daniele Vilone, Eugenia Polizzi.

**Writing – original draft:** Daniele Vilone, Eugenia Polizzi.

**Writing – review & editing:** Daniele Vilone, Eugenia Polizzi.

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
