## [Decision Letter · Decision Letter 0]

20 Sep 2023

PONE-D-23-21945Modelling opinion misperception and the emergence of silence in online social systemPLOS ONE

Dear Dr. Vilone,

Thank you for submitting your manuscript to PLOS ONE. After careful consideration, we feel that it has merit but does not fully meet PLOS ONE’s publication criteria as it currently stands. Therefore, we invite you to submit a revised version of the manuscript that addresses the points raised during the review process.

The outcome of the peer-review process for your paper is critical, as two expert reviewers have evaluated the manuscript, but one of them has raised significant concerns. I recommend that the authors carefully revise their manuscript in light of these comments. ==============================

We look forward to receiving your revised manuscript.

Kind regards,

Pierluigi Vellucci

Academic Editor

PLOS ONE

“Daniele Vilone was partially supported by project SERICS (PE00000014) under the MUR National Recovery and Resilience Plan funded by the European Union  NextGenerationEU. Eugenia Polizzi was partially supported by the EU H2020 ICT48 project "Humane AI Net" under contract n 952026.”

“This work was partially supported by project SERICS (PE00000014) under the MUR

National Recovery and Resilience Plan funded by the European Union

NextGenerationEU and by the EU H2020 ICT48 project ”Humane AI Net” under

contract n 952026.”

“Daniele Vilone was partially supported by project SERICS (PE00000014) under the MUR National Recovery and Resilience Plan funded by the European Union  NextGenerationEU. Eugenia Polizzi was partially supported by the EU H2020 ICT48 project "Humane AI Net" under contract n 952026.”

6. We note that you have indicated that data from this study are available upon request. PLOS only allows data to be available upon request if there are legal or ethical restrictions on sharing data publicly. For more information on unacceptable data access restrictions, please see http://journals.plos.org/plosone/s/data-availability#loc-unacceptable-data-access-restrictions.

Reviewers' comments:

Reviewer's Responses to Questions

**Comments to the Author**

1. Is the manuscript technically sound, and do the data support the conclusions?

Reviewer #1: Yes

Reviewer #2: Partly

2. Has the statistical analysis been performed appropriately and rigorously? 

Reviewer #1: Yes

Reviewer #2: N/A

3. Have the authors made all data underlying the findings in their manuscript fully available?

Reviewer #1: No

Reviewer #2: Yes

4. Is the manuscript presented in an intelligible fashion and written in standard English?

Reviewer #1: No

Reviewer #2: Yes

5. Review Comments to the Author

Reviewer #1: This study describes (computer) simulation experiments to explore different theoretical pathways that can explain the empirical finding of “misperception” – that is, humans’ erroneous impression of how a certain opinion or belief is distributed in the population. I list here my comments – I put them together by theme, generally trying to list the more general ones at the top and leaving the more secondary ones at the bottom.

On the substantive level, I invite the authors to reflect on the limitations of their study, particularly on how dependent their results are on key modeling assumptions. I will make three examples. My recommendation is to surface and motivate these assumptions upfront in the manuscript (e.g. where the model is introduced) or at least to surface and comment on them in a wider discussion on the limitations and generalizability of the study at the end of the manuscript.

- One of the unappreciated key assumptions, I believe, is that privately held beliefs (denoted b) are fixed. This implies that voiced opinions can change, whereas private opinions do not (or, if they do, they can change at a much slower rate such that it is not captured by the model). As far as I know this assumption is quite unusual in the literature and needs to be motivated. It seems reasonable to expect that results would be different if individuals adjusted their private beliefs, too: I say this with an eye on literature on behavioral mechanisms such as conformity and cognitive dissonance, where human tendency to conform with the perceived norm is not only superficial (read: it does not only affect publicly voiced opinions), but it may actually affect one’s privately held belief, too. If so, I would expect misperception to be a less prominent emerging feature of the model.

- Another key assumption is about the social influence mechanism implemented in the model. Page 4 line 122 introduces what is essentially a “majority rule” model where the object of influence is not agents’ privately-held opinion but the opinion they opt to voice. Different social influence mechanisms are likely to produce different macro-level outcomes. Readers should be made aware that these results only apply to the extent that individuals’ choice as to which opinion to express follows a majority rule mechanism. (Incidentally, the authors are also advised to cite the relevant literature on the majority rule model – currently lacking).

- There is another assumption that might deserve some consideration – but which might be less important for this study and more important for future work: the granularity/dimensionality of the opinion scale. Is it reasonable to expect results to change if the opinion scale had more levels than two? And if it was multidimensional (e.g. if individuals simultaneously had more beliefs about related things and could decide which one to voice when interrogated?)

Another general comment I have – weighing negatively in my evaluation of the manuscript – concerns the form. The most problematic issue in this regard is that key concepts are lacking a proper definition and are referred to using different inconsistent labels. This makes the manuscript difficult to understand where it matters most. I make here some examples – note that the examples I chose are particularly delicate in that they involve some main concepts in the study whose meaning should be unambiguous.

- Behavior / decision / opinion / position / belief / information: these terms seem to be used interchangeably in the introduction, to the point that it is not clear if and when the use of one instead of the other is meaningful.

- Contrarian / minoritarian agent seem to mean the same thing. Furthermore, not only the label is inconsistent: the underlying definition of contrarian is problematic. On page 2, line 54 contrarians are defined as “individuals holding extreme positions”. Opinions in this study are modeled as essentially binary. How can an opinion on a binary scale be “extreme”? In a binary opinion scale all agents will have an opinion at the extreme end of the scale, so they all will be “holding extreme positions”. Note that on page 5, line 162 a new definition of “contrarians” is introduced: “committed users holding unpopular opinions”. This works better, but it still isn’t perfect: what does “committed” mean in this context, and is it really necessary for this definition? Furthermore, immediately following the new definition (line 165) from the implementation of contrarians in the model it becomes apparent that the essence of being a contrarian is not so much about the opinion being held, but the consistency with which contrarians voice their true opinion (opinion strength is set to 1 for contrarians).

- Misperception / social bias seem to mean the same thing. If not, the first occurrence of “social bias” (line 23) makes the meaning of this paragraph obscure. Incidentally, the sentence on lines 22-24 has other issues. It is also unclear what “online social system” refers to; and the sentence tries to introduce too many ideas at once: (1) that there is empirical evidence of misperception; (2) that there is some theoretical puzzle concerning the empirically-observed misperception; (3) that misperception -- and/or the theoretical puzzle about it -- involves online social networks. If I understand correctly, this sentence can be rewritten with less ambiguity as follows: "Despite empirical evidence of misperceptions about the popularity of certain behaviors and opinions -- particularly in online social networks [refs], -- we still lack a clear understanding of how these misperceptions emerge in the first place. "

- Opinion expression / opinion declaration / and other various circumlocutory expressions also seem to mean the same thing.

Other more secondary comments:

- Plos One submission guidelines mandate, with few exemptions, the publication of research data in public repositories. The data statement for the manuscript seems insufficient in this regard, as replication scripts are made available upon request to the authors whereas they should be deposited to a public repository.

- About the first equation (page 10): square brackets are here used to signify some operation on the sum of w over all neighbors of i. I am clueless as to what that operation is (the mode perhaps?). Please check for correctness and consider adding explanatory text.

- Line 64: This is the first occurrence of the terms “exogenous” and “endogenous” to refer respectively to illusion of majority and pluralistic ignorance. However, I do not get why one would be exogenous and the other endogenous. I can understand they are different phenomena -- one is a feature of social networks and the other is a social phenomenon -- but I don't clearly see how endo- or exogeneity captures the essence of their difference.

- About Figures. Some figures are very busy / information dense. Information density is not a problem, but it requires holding the reader by hand. Labelling all axes is a must. High(er) pixel density/resolution will be necessary. Using colorblind friendly color palettes is recommended, particularly when the line style or pointer shape do not disambiguate between elements (avoid e.g. purple vs black in Figure 1 right; and red vs green in Figure 2). Making sure that figure captions are close to the actual pictures is a welcome help to reviewers.

About figure 1 specifically: is there a reason why misperception and public vs private mismatch are only highlighted for the black lines? If the only reason is that purple lines do not allow for enough space, perhaps the highlight in the plot can be replaced with an explanation in-text or in caption. And on the right panel, the X-axis label reads “K_0”. Should it not be “K_c”?

- Line 30: “support the role of” -> perhaps you mean “confirm the importance of”

- Line 35: what is meant by "understanding of the origin of the information"? Do you mean "understanding of the truthfulness of the information", "trustworthiness of the person who is sharing the information", "whether the information is truly held/believed by many and not just very minoritarian", or something else?

- Line 37: “Observational data […]”. I can understand what you mean because I know that there is a literature on opinion dynamics and I agree that it can be used to model the emergence/spread of misperception and ultimately polarization. However, the generalist reader will be clueless at this point. You mention behavioral changes and complex dynamics, but the introduction at this point has only mentioned phenomena such as sharing/receiving information/opinions and estimating how widespread in society such information/opinion is. These phenomena per se do not involve any behavior change or complex dynamics. Therefore, it is not clear at this point what dynamics the text is referring to.

- Lines74-75: “[…] when a committed minority […]”. This sentence is unclear. Please check for syntax. What I understand: "when a community of agents -- agreeing, to varying degrees, on a moderate opinion -- is joined by a small group of contrarians."

- Line 94. I do not understand what the word “heterogeneous” means here.

- Line 106: “[…] σ_i: this is the only variable describing the internal state of the agents”. Is b_i -- the private opinion – not also an internal state of agents?

- Line 111: “see below for clarifications”. I have not seen or recognized what clarification this is referring to. The first concept that I think relates to w_i=0 is silence – which comes into play much later in the manuscript.

- Line 176: “prestige”. How prestige relates to degree requires a bit of explanation. An inexpensive solution perhaps is to say “centrality” instead?

Reviewer #2: This paper addresses the investigation of opinion misperception in the context of social media, a phenomenon exacerbated by the silence of a small yet highly influential group of users.

Numerical simulations are shown to highlight the influence of the connections structure on the distortion of the perception of individuals' belief systems.

Given the substantial impact of this phenomenon on opinion dynamics, it is a subject worthy of investigation. The paper may be considered for publication after addressing the following points:

1- Pag.2: In the Introduction some crucial literature is missing. For example the concept of contrarian agent is introduced without

taking into account the works done in this framework by Serge Galam and his collaborators.

Moreover the authors say regarding the ABM models that "they are less suited to examine the processes by which a samll fraction of individuals holding extreme positions (hereby "contrarians") can fuel opinion misperception...". In this framework, I would suggest to check and compare the following papers:

- Galam, Serge. "Contrarian deterministic effects on opinion dynamics:“the hung elections scenario”." Physica A: Statistical Mechanics and its Applications 333 (2004): 453-460.

- Borghesi, Christian, and Serge Galam. "Chaotic, staggered, and polarized dynamics in opinion forming: The contrarian effect." Physical Review E 73.6 (2006): 066118.

- Jacobs, Frans, and Serge Galam. "Two-opinions-dynamics generated by inflexibles and non-contrarian and contrarian floaters." Advances in Complex Systems 22.04 (2019): 1950008.

- Galam, Serge. "From 2000 Bush–Gore to 2006 Italian elections: voting at fifty-fifty and the contrarian effect." Quality & quantity 41.4 (2007): 579-589.

- Javarone, Marco Alberto. "Social influences in opinion dynamics: the role of conformity." Physica A: Statistical Mechanics and its Applications 414 (2014): 19-30.

- Nyczka, Piotr, and Katarzyna Sznajd-Weron. "Anticonformity or independence?—Insights from statistical physics." Journal of Statistical Physics 151.1-2 (2013): 174-202.

- Iacomini, Elisa, and Pierluigi Vellucci. "Contrarian effect in opinion forming: insights from Greta Thunberg phenomenon." The Journal of Mathematical Sociology 47.2 (2023): 123-169.

However, misperception is a different "topic" from contrarians agents.

2- Pag.2: Typo in the last sentence: "simulation works"

3- Pag.3: The first assumption of the model is b_i=+-1, constant in time. Does it mean that no one will change its opinion?

4- Pag.5: The definition of contrarian agent does not fit with the definition in the literature. Maybe it is more appropriate to call them committed agents, or as the authors also suggested, "hard-cores". The main difference is that a contrarian agent has the opinion which is the opposite to a specific agent/group/majority, depending on which type of contrarian is considered, but it does not influence the strength of the belief.

5- Pag.6: In order to improve the readability of the paper, the figures should be placed throught the text, above the caption. Moreover, the plots are blurred.

6- Pag.11: As a curiosity, do you think that the spiral of silence can be seen from the data?

6. PLOS authors have the option to publish the peer review history of their article (what does this mean?). If published, this will include your full peer review and any attached files.

Reviewer #1: No

Reviewer #2: No

---

## [Author Response · Author response to Decision Letter 0]

9 Nov 2023

See file attached with the responses to each reviewers' points ("VP2023_Responses2reviewers").

---

## [Decision Letter · Decision Letter 1]

6 Dec 2023

Modeling opinion misperception and the emergence of silence in online social system

PONE-D-23-21945R1

Dear Dr. Vilone,

We’re pleased to inform you that your manuscript has been judged scientifically suitable for publication and will be formally accepted for publication once it meets all outstanding technical requirements.

Kind regards,

Pierluigi Vellucci

Academic Editor

PLOS ONE

Additional Editor Comments (optional):

Reviewers' comments:

Reviewer's Responses to Questions

**Comments to the Author**

1. If the authors have adequately addressed your comments raised in a previous round of review and you feel that this manuscript is now acceptable for publication, you may indicate that here to bypass the “Comments to the Author” section, enter your conflict of interest statement in the “Confidential to Editor” section, and submit your "Accept" recommendation.

Reviewer #1: (No Response)

Reviewer #2: All comments have been addressed

2. Is the manuscript technically sound, and do the data support the conclusions?

Reviewer #1: Yes

Reviewer #2: (No Response)

3. Has the statistical analysis been performed appropriately and rigorously? 

Reviewer #1: Yes

Reviewer #2: (No Response)

4. Have the authors made all data underlying the findings in their manuscript fully available?

Reviewer #1: No

Reviewer #2: (No Response)

5. Is the manuscript presented in an intelligible fashion and written in standard English?

Reviewer #1: Yes

Reviewer #2: Yes

6. Review Comments to the Author

Reviewer #1: My compliments to the authors for the thorough revision of their article. My comments and suggestions were generally satisfactorily addressed and/or responded to. There is only one partial exception, which hopefully will be easy to remedy, and I have a short list of what I think are simply spurious typos.

The partial exception concerns the data sharing – which, in this case, entails sharing the simulation scripts. The scripts were uploaded as supplementary material alongside the article, but the article is not yet fully complying with the “FAIR” principles that PLOS strives for (https://journals.plos.org/plosone/s/data-availability), nor with the specific requirements defined in the PLOS data policy (see in particular the provisions concerning sharing software at http://www.plosone.org/static/policies.action#sharing). Actual and potential issues include:

- The scripts do not seem to be complete: for example, upon quick look I could not find the code that generated the plots reproduced in the article.

- Scripts are not commented (or at least they are not commented in English).

- Documentation is missing – including a description of what software (and software version) the scripts are written for; a description of what script does what; and/or some user instructions as to how to run the scripts to reproduce the results in the article.

- Though perhaps not a strict requirement, the PLOS data policy document further advises depositing the code to an external repository, licensing it, and crosslinking the article and the code repository.

Lastly, the spurious typos I think I found:

- New Figure 1 (bottom panel). See the text label added to the top-left corner of the plot area. Should “K_0” not be replaced with “K_c”?

- New Figure 4, Y-axis label. Should “contrarian” not be replaced with “hard-core agent” or similar?

- Line 83: replace “this” with “these”.

- Line 98: “e.g., illusion of majority”. Should it not be “i.e., illusion of majority”?

I hope all this can be easily amended and I hope to see the article published soon.

Reviewer #2: The authors addressed my comments justifying and motivating their answers with clarity and precision.

I just recommend an additional reading to address the remaining typos in the manuscript.

7. PLOS authors have the option to publish the peer review history of their article (what does this mean?). If published, this will include your full peer review and any attached files.

Reviewer #1: No

Reviewer #2: No

---

## [Editor Report · Acceptance letter]

14 Dec 2023

PONE-D-23-21945R1 

PLOS ONE

Dear Dr. Vilone, 

I'm pleased to inform you that your manuscript has been deemed suitable for publication in PLOS ONE. Congratulations! Your manuscript is now being handed over to our production team.

Kind regards, 

on behalf of

Dr. Pierluigi Vellucci 

Academic Editor

PLOS ONE